# Afatinib and Dacomitinib Efficacy, Safety, Progression Patterns, and Resistance Mechanisms in Patients with Non-Small Cell Lung Cancer Carrying Uncommon *EGFR* Mutations: A Comparative Cohort Study in China (AFANDA Study)

**DOI:** 10.3390/cancers14215307

**Published:** 2022-10-28

**Authors:** Hong-Shuai Li, Shou-Zheng Wang, Hai-Yan Xu, Xiang Yan, Jin-Yao Zhang, Si-Yu Lei, Teng Li, Xue-Zhi Hao, Tao Zhang, Guang-Jian Yang, Li-Qiang Zhou, Peng Liu, Yu-Ying Wang, Xing-Sheng Hu, Pu-Yuan Xing, Yan Wang

**Affiliations:** 1Department of Medical Oncology, National Cancer Center/National Clinical Research Center for Cancer/Cancer Hospital, Chinese Academy of Medical Sciences and Peking Union Medical College, Beijing 100021, China; 2Department of Medical Oncology, Beijing Chest Hospital, Beijing Tuberculosis and Thoracic Tumor Research Institute, Capital Medical University, Beijing 101149, China; 3Department of Comprehensive Oncology, National Cancer Center/National Clinical Research Center for Cancer/Cancer Hospital, Chinese Academy of Medical Sciences and Peking Union Medical College, Beijing 100021, China; 4Department of Oncology, The Fifth Medical Center, Chinese PLA General Hospital, Beijing 100000, China; 5Department of Radiotherapy, National Cancer Center/National Clinical Research Center for Cancer/Cancer Hospital, Chinese Academy of Medical Sciences and Peking Union Medical College, Beijing 100000, China; 6Department of Respiratory Medicine, Shandong Cancer Hospital and Institute, Shandong First Medical University and Shandong Academy of Medical Sciences, Ji’nan 250000, China

**Keywords:** non-small cell lung cancer, dacomitinib, afatinib, uncommon EGFR mutations, efficacy, safety, progression patterns, resistance

## Abstract

**Simple Summary:**

Afatinib has been approved for patients with lung cancer carrying uncommon epidermal growth factor receptor gene (*EGFR*) mutations. Dacomitinib, another second-generation inhibitor, has also shown promising potential for these mutations. This is the first and largest comparative study on second-generation inhibitors in patients with uncommon *EGFR* mutations to date. We found that dacomitinib demonstrated a more favorable activity with manageable toxicity compared with afatinib, which provided more evidence for dacomitinib application in this setting.

**Abstract:**

(1) Background: Afatinib has been approved for patients with non-small cell lung cancer (NSCLC) carrying major uncommon epidermal growth factor receptor gene (*EGFR*) mutations. Dacomitinib, another second-generation tyrosine kinase inhibitor, has also shown promising potential for uncommon *EGFR* mutations. However, no comparative study has been conducted. (2) Methods: Two cohorts were employed: the AFANDA cohort, an ambispective cohort including 121 patients with uncommon *EGFR* mutations admitted to two tertiary hospitals in China, and an external validation afatinib cohort (ex-AC), extracted from the Afatinib Uncommon *EGFR* Mutations Database (N = 1140). The AFANDA cohort was divided into an afatinib cohort (AC) and a dacomitinib cohort (DC) for internal exploration. Objective response rate (ORR), progression-free survival (PFS), and adverse events (AEs) were assessed for comparison. Progression patterns and resistance mechanisms were explored. (3) Results: In total, 286 patients with advanced NSCLC carrying uncommon *EGFR* mutations treated with afatinib or dacomitinib were enrolled, including 79 in the AFANDA cohort (44 in the DC, 35 in the AC) and 207 in the ex-AC. In internal exploration, the ORR of the DC was significantly higher than that of the AC (60.5 vs. 26.7%, *p* = 0.008), but there was no significant difference in median PFS between the DC and the AC (12.0 months vs. 10.0 months, *p* = 0.305). Multivariate analysis confirmed an independent favorable effect of dacomitinib on PFS (hazard ratio (HR), 1.909; *p* = 0.047). In external validation, multivariate analysis confirmed the independent prognostic role of dacomitinib in PFS (HR, 1.953; *p* = 0.029). Propensity score matching analysis confirmed the superiority of dacomitinib over afatinib in terms of PFS in both univariate and multivariate analyses. Toxicity profiling analysis suggested more G1 (*p* = 0.006), but fewer G3 (*p* = 0.036) AEs in the DC than in the AC. Progression patterns revealed that the incidence of intracranial progression in the AC was significantly higher than that in the DC (50 vs. 21.1%, *p* = 0.002). Drug resistance analysis indicated no significant difference in the occurrence of T790M between the AC and the DC (11.8 vs. 15.4%, *p* = 0.772). (4) Conclusions: Compared with afatinib, dacomitinib demonstrated a more favorable activity with manageable toxicity and different progression patterns in patients with NSCLC carrying uncommon *EGFR* mutations.

## 1. Background

Lung cancer remains one of the most prevalent and deadly cancers worldwide [1,2]. Targeted therapies, represented by epidermal growth factor receptor-tyrosine kinase inhibitors (EGFR-TKIs), have greatly improved the quality of life and prognosis of patients with non-small cell lung cancer (NSCLC) [2]. Patients carrying common *EGFR* mutations (including in-frame deletion mutations in exon 19 (19del) and missense mutations in exon 21 (L858R)) significantly benefit from EGFR-TKIs; however, because of the high heterogeneity of uncommon *EGFR* mutations (defined as mutations other than classical mutations), patients carrying such mutations differ dramatically in their sensitivity to TKIs [3,4,5,6,7]. In general, patients carrying uncommon *EGFR* mutations are less sensitive to TKIs than those harboring classical mutations and tend to have an unfavorable prognosis [8].

Studies by Yang et al. have demonstrated the excellent efficacy of afatinib in patients with NSCLC carrying major uncommon mutations (including G719X, S768I, and L861Q) [9,10]. Afatinib has been approved by the Food and Drug Administration (FDA) and is endorsed by the National Comprehensive Cancer Network guidelines as the preferred first-line treatment in this subset of patients. Dacomitinib, another highly selective, irreversible second-generation (2G) EGFR-TKI, has been approved as a first-line treatment in patients with NSCLC carrying classical *EGFR* mutations as it significantly improved progression-free survival (PFS) as compared with gefitinib in the ARCHER 1050 study [11]. In this context, it is desirable to assess the therapeutic potential of dacomitinib in patients carrying uncommon mutations. In fact, dacomitinib has shown potential for the treatment of NSCLC in patients harboring uncommon mutations [12,13,14,15,16,17]. We previously demonstrated the efficacy of dacomitinib in patients with NSCLC carrying uncommon *EGFR* mutations in first-line and later-line settings [16,17,18].

Based on our previous findings, in this study, we compared dacomitinib and afatinib in terms of efficacy, safety, progression patterns, and resistance mechanisms in patients with NSCLC carrying uncommon *EGFR* mutations with the aim of obtaining valuable evidence for clinical decision making.

## 2. Methods

### 2.1. Study Design and Data Resources

This was a comparative cohort study in two centers in China. Patients with advanced or recurrent NSCLC carrying uncommon *EGFR* mutations treated with afatinib or dacomitinib were eligible for evaluation. Two cohorts were employed in the current study. The AFANDA cohort, an ambispective cohort, was used for internal exploration. It included 121 patients carrying uncommon *EGFR* mutations who received afatinib or dacomitinib between 1 January 2017 and 15 April 2022 at two tertiary hospitals in China, namely, the Chinese PLA Hospital and the Chinese National Cancer Center. The external validation afatinib cohort (ex-AC) was extracted from the publicly available Afatinib Uncommon *EGFR* Mutations Database (https://www.uncommonegfrmutations.com/) (last extraction date: 20 June 2022), which includes 1140 patients harboring uncommon *EGFR* mutations treated with afatinib retrieved from publications [10,19,20,21,22,23,24,25,26]. The AFANDA cohort was divided into an afatinib cohort (AC) and a dacomitinib cohort (DC) for internal exploration, and the ex-AC was employed for external validation.

### 2.2. Patient Selection Criteria

The patient selection criteria were as follows. For the AFANDA cohort: (1) patients with histologically or cytologically confirmed diagnosis of advanced or recurrent NSCLC; (2) patients harboring uncommon *EGFR* mutations (other than T790M, exon 20 insertions (EX20ins), and compound 19del/L858R); (3) patients from whom tumor tissue or cell-free DNA from humoral samples (including plasma, cerebrospinal fluid, and pleural effusion) before dacomitinib/afatinib administration were analyzed utilizing next-generation sequencing (NGS) or amplification refractory mutation system PCR at the above two centers or at a CAP-accredited institution; (4) patients who received dacomitinib or afatinib monotherapy; and (5) patients whose survival data were complete. For the ex-AC: (1) patients from clinical trials/clinical studies other than case studies/case series and the compassionate-use program (CUP)/expanded-access program (EAP); (2) patients harboring uncommon *EGFR* mutations (other than T790M, EX20ins, and compound 19del/L858R); and (3) patients whose survival data were complete.

### 2.3. Treatment and Efficacy/Toxicity Evaluation

All enrolled patients received single-agent dacomitinib or afatinib. The dacomitinib or afatinib dosage was determined according to the patient’s comorbidities, weight, and physical status. The objective response, including complete response (CR) and partial response (PR), and disease control, including CR, PR, and stable disease (SD), were judged according to the RECIST (version 1.1) guideline. Toxicity was assessed according to the CTCAE 5.0 criteria.

### 2.4. Exploration of Resistance Mechanisms 

Cell-free DNA from humoral samples (including plasma, cerebrospinal fluid, and pleural effusion) after dacomitinib/afatinib resistance were analyzed utilizing next-generation sequencing (NGS) using an ultra-deep (20,000×) 168-gene panel named LungPlasma (Burning Rock Biotech, Guangzhou, China). The analyses were performed at the above two centers or at a CAP-accredited institution (commercially, >200-gene panel).

### 2.5. Statistical Analysis

The study cutoff date was 15 May 2022. PFS was defined as the interval from dacomitinib or afatinib administration to disease progression or death due to any cause. Patients lost to follow-up were reviewed, and the last determinable survival time was taken as the end of follow-up.

Categorical data are reported as numbers and percentages and were analyzed using the chi-square test. Survival analysis was conducted using the Kaplan–Meier method. Univariate and multivariate Cox proportional hazards models were utilized to assess the associations between variables and PFS. The multivariate model included variables that were found to be significant in the univariate analyses and variables that were considered clinically significant.

Propensity score matching (PSM), applying a caliper width of 0.2 of the standard deviation, was conducted to produce matched groups of the DC and the ex-AC. The analysis was based on age, sex, smoking status, treatment history, brain metastases status, mutation category, exon category, and compound mutation status. SPSS software (version 23.0, IBM Corporation, Armonk, NY, USA) was employed to conduct the PSM analysis [27,28].

All statistical analyses were conducted using R software (version 4.0.0, R Foundation). A two-sided *p*-value < 0.05 was considered statistically significant.

## 3. Results

A flow chart of patient selection is shown in Figure 1. In total, 286 patients were enrolled, including 79 in the AFANDA cohort (44 in the DC, 35 in the AC) and 207 in the ex-AC [9,10,19,20,21,22,23,24,25,26]. The median follow-up time was 12.4 months (range: 0.8–51.2 months) in the AFANDA cohort and 32.2 months (0–70.8 months) in the ex-AC.

### 3.1. Baseline Characteristics in the DC and the AC

The clinical, pathological, and molecular characteristics of patients in the DC and the AC are provided in Table 1. All characteristics were balanced and comparable between the two cohorts. More than 65% of the patients were female and never-smokers, and more than 25% had brain metastases. Most patients in both cohorts had a performance status (Eastern Cooperative Oncology Group (ECOG) PS) of 1. G719X was the most common mutation in both cohorts, followed by L861Q and S768I in the DC and S768I and L861Q in the AC. Uncommon mutations most frequently involved exon 18. The detailed mutation landscapes of the DC and the AC are shown in Appendix A. Dacomitinib and afatinib were administrated as the first-line therapy in more than 60% of patients.

### 3.2. Efficacy Evaluation, Survival Analysis, and Subgroup Analysis in the DC and the AC

The objective response rate (ORR) was significantly higher in the DC than in the AC (60.5 vs. 26.7%, *p* = 0.008) (Figure 2a). The Kaplan–Meier analysis revealed a longer, albeit not significantly longer, median PFS (mPFS) in the DC than in the AC (12.0 months vs. 10.0 months, *p* = 0.305) (Figure 2b). The ORRs of subtypes, including mutation category (Figure 2c) and exon category (Figure 2d), were generally higher in the DC than in the AC, especially “other mutation types” in the mutation category (66.7 vs. 9.1%, *p* = 0.003). Subgroup analysis suggested that patients in the AC tended to have a higher risk of progression than those in the DC in most subgroups, particularly for other mutation types in the mutation category (HR, 2.844; *p* = 0.040) (Figure 2e).

### 3.3. Univariate and Multivariate Analyses in the Pooled DC and AC

To determine the effects of the different interventions and characteristics on PFS, we pooled the DC and the AC and then conducted univariate and multivariate analyses. The univariate analysis revealed that PFS was significantly associated with brain metastases (*p* = 0.004), tumor burden (*p* < 0.001), ECOG PS (*p* < 0.001), and application line (*p* = 0.018) (Table 2). The multivariate analysis confirmed an independent favorable prognostic role of dacomitinib in PFS (DC vs. AC: HR, 1.909; 95% CI, 1.009–3.610; *p* = 0.047), and ECOG PS (*p* < 0.001) and application line (*p* = 0.043) were also independent factors in the final regression model (Table 2).

### 3.4. Toxicity Analysis in the DC and the AC

The DC and the AC shared a spectrum of AEs, which mainly included rash, diarrhea, oral mucositis, paronychia, and dry skin (Table 3). The main AEs in the DC were rash (90.9%), diarrhea (77.3%), and oral mucositis (61.4%), whereas in the AC, the main AEs were diarrhea (62.9%), oral mucositis (57.1%), and rash (51.4%). The proportion of G1 AEs in the DC was significantly higher than that in the AC (*p* = 0.006), whereas the proportion of G3 diarrhea in the AC was significantly higher than that in the DC (*p* = 0.036).

### 3.5. External Validation in the DC and the ex-AC

The clinical, pathological, and molecular characteristics of patients in the DC and the ex-AC are provided in Table 4. Most characteristics were significantly different between the two cohorts. The detailed mutation landscape of the ex-AC is shown in Appendix A.

No significant differences were observed regarding the ORRs (Figure 3a) and the mPFS (Figure 3b) between the DC and the ex-AC (ORRs: 60.5 vs. 51.1%, *p* = 0.472; mPFS: 12.0 months vs. 10.4 months, *p* = 0.325). The ORRs of subtypes, including mutation category (Figure 3c) and exon category (Figure 3d), did not significantly differ. Subgroup analysis suggested that patients in the ex-AC had a greater risk of progression than those in the DC in most subgroups, particularly in the no brain metastases (HR, 2.258; *p* = 0.036) and TKI-naïve (HR, 2.415; *p* = 0.023) subgroups (Figure 3e).

Univariate analysis of the pooled DC and ex-AC indicated that sex (*p* = 0.035) and treatment history (*p* = 0.033) were significantly associated with PFS. To reduce the confounding effects of imbalances in the baseline characteristics, a multivariate analysis was subsequently conducted, and this confirmed the independent favorable prognostic role of dacomitinib in PFS (DC vs. ex-AC: HR, 1.953; 95% CI, 1.071–3.563; *p* = 0.029) (Table 5).

### 3.6. External Validation in the DC and the ex-AC after PSM

Most characteristics, except age, were not balanced between the DC and the ex-AC, as indicated in Table 4. To reduce the confounding effects of imbalances in the baseline characteristics, a 1:2 matching (one case from the DC to two cases from the ex-AC) PSM based on all characteristics was conducted to create matched groups of the DC and the ex-AC, which resulted in a good balance of all selected variables (Table 4, Appendix A). A comparison of the characteristics between the DC (N = 28) and the ex-AC (N = 48) after the PSM is provided in Table 4.

The ORR was higher, albeit not significantly higher, in the DC than in the ex-AC (74.1 vs. 51.1%, *p* = 0.211) (Figure 4a). The mPFS was significantly longer in the DC than in the ex-AC (15.9 months vs. 10.4 months, *p* = 0.043) (Figure 4b). Consistent with the results before PSM, no significant differences in the ORR were observed among the mutation categories (Figure 4c) and exon categories (Figure 4d). Subgroup analysis suggested that patients in the DC had a lower risk of progression than those in the ex-AC in most subgroups, particularly in the no brain metastases (HR, 0.443; *p* = 0.076) and TKI-naïve (HR, 0.355; *p* = 0.033) subgroups (Figure 4e).

Univariate analysis of the pooled DC and ex-AC indicated that only dacomitinib intervention (*p* = 0.033) was significantly associated with PFS, and multivariate analysis confirmed the independent favorable prognostic role of dacomitinib in PFS (DC vs. ex-AC: HR, 3.030; 95% CI, 1.304–7.042; *p* = 0.010) (Appendix A).

### 3.7. Progression Patterns, Resistance Mechanisms, and Subsequent Therapies

Analysis of the progression patterns of patients (N = 45) (Figure 5a,b) revealed a significantly higher incidence of intracranial progression in the AC than in the DC (50% vs. 21.1%, *p* = 0.002) (Appendix A). Liquid biopsy-based NGS data were available for 30 patients who developed drug resistance. Drug resistance analysis indicated no significant difference in the occurrence of secondary T790M between the AC and the DC (11.8 vs. 15.4%, *p* = 0.772) (Figure 5c) (Appendix A). The top three off-target mutations were identified as *TP53* (53%), *LRP1B* (13%), and *PTEN* (13%) in the pooled cohort, as *TP53* (46.2%), *PTEN* (38.5%), and *PKHD1* (38.5%) in the DC, and as *TP53* (58.8%), *LRP1B* (17.6%), and *CHD8* (17.6%) in the AC (Figure 5c). The bridging therapies used after the onset of dacomitinib and afatinib resistance in the DC and the AC, respectively, are summarized in Figure 5d,e. Combined application of dacomitinib and an anti-angiogenic agent (AAA) (bevacizumab or anlotinib) was the most frequently used (25%) in the DC after drug resistance, whereas afatinib + AAA ± brain radiotherapy was the most frequently used (24%) in the AC. The proportion of patients receiving local interventions (including brain radiotherapy and local interventional therapies) was significantly lower in the DC than in the AC (10 vs. 56%, *p* = 0.001) (Appendix A).

## 4. Discussion

In the current study, we tentatively compared 2G TKIs in patients with NSCLC carrying uncommon *EGFR* mutations with the aim of obtaining valuable evidence for clinical reference. The findings of the internal cohort exploration and external cohort validation indicated that dacomitinib had an efficacy advantage over afatinib in patients with NSCLC carrying uncommon *EGFR* mutations with a manageable safety profile, which warrants further clinical exploration and confirmation. In addition, we provided data on progression patterns and resistance mechanisms in patients treated with 2G TKIs. To the best of our knowledge, this is the first and largest comparative study to date on 2G TKIs in patients with NSCLC carrying uncommon *EGFR* mutations.

Previous evidence has demonstrated that, compared with 1G TKIs, the 2G TKI afatinib has a more favorable effect on PFS in patients with NSCLC harboring uncommon *EGFR* mutations (afatinib vs. gefitinib vs. erlotinib: 10.5 months vs. 3.0 months vs. 0.9 months, *p* = 0.013) [29]. A post-hoc analysis conducted by Yang et al. revealed that for patients carrying the major uncommon mutations G719X, S768I, and L861Q, the ORRs and mPFS reached 77.8, 100, and 56.3% and 13.8 months, 14.7 months, and 8.2 months, respectively [9]. Therefore, in 2018, afatinib was approved by the FDA for patients with NSCLC harboring major uncommon mutations. The 3G TKI osimertinib has also been shown to be efficacious in patients harboring G719X, S768I, and L861Q, with ORRs of 53, 38, and 78% and an mPFS of 8.2 months, 12.3 months, and 15.2 months, respectively, in the KCSG-LU15-09 study [30]. At present, data on the treatment of uncommon *EGFR* mutations with dacomitinib are limited, and most are from case reports and small retrospective clinical studies. Previously published studies in patients with NSCLC carrying uncommon *EGFR* mutations treated with dacomitinib are summarized in Appendix A.

In our previous studies, we successively demonstrated the promising activity of dacomitinib in patients with NSCLC carrying uncommon *EGFR* mutations in first-line and later-line settings and the manageable safety profile of dacomitinib [16,17,18]. In the first-line setting [17], dacomitinib achieved an ORR of 72.2% and a disease control rate of 100% in patients carrying major uncommon *EGFR* mutations, and the mPFS and overall survival were not reached. Specifically, the ORRs in patients harboring G719X, L861Q, and S768I were 66.7%, 50%, and 100%, respectively, and the mPFS was not reached in any of the three mutation groups, which results were in line with the findings of the post-hoc analysis of afatinib in the LUX-Lung series of trials [9]. We included not only the major uncommon *EGFR* mutations, but also other uncommon mutations (besides T790M, EX20ins, and compound 19del/L858R) in the current study. In this setting, dacomitinib obtained an ORR of 60.5%, a disease control rate of 90.7%, and an mPFS of 12.0 months. Specifically, in patients carrying G719X, L861Q, S768I, and other uncommon mutations, the ORRs were 62.5, 54.5, 66.7, and 66.7% and the mPFS was 12 months, unreached, 6.5 months, and 12.5 months, respectively. This suggests that dacomitinib also has therapeutic potential for non-major uncommon *EGFR* mutations [14,16,31,32].

The secondary T790M mutation rate in the patients in our study (13%) was substantially lower than those reported in patients harboring classical mutations (40.7–64.3%) [33,34], which result was consistent with the findings reported by Wu et al. [34] and Lin et al. [35]. In the study by Wu et al., patients with uncommon *EGFR* mutations receiving afatinib had a lower secondary T790M mutation rate than those harboring classical mutations (64.3% for 19del, 45.5% for L858R, and only 16.7% for uncommon *EGFR* mutations, *p* = 0.142) [34]. The study by Lin et al. revealed that an uncommon *EGFR* mutation was a negative independent indicator for secondary T790M mutation (adjusted odds ratio 0.14, 95% CI, 0.02–0.97; compared with 19del, *p* = 0.047). Furthermore, we did not observe a significant difference in the incidence of secondary T790M between the DC and the AC. However, considering the small cohort size in this study, a firm conclusion cannot be drawn. In the context of uncommon mutations, the efficacy of sequential osimertinib after 2G TKI resistance development remains to be further explored.

Studies have demonstrated that dacomitinib has a potent efficacy in patients with NSCLC carrying brain metastases [36,37], and this has been specifically confirmed in patients with uncommon *EGFR* mutations [17]. In line herewith, we found a significantly lower incidence of intracranial progression in the DC than in the AC (21.1 vs. 50%, *p* = 0.002). Nevertheless, we did not observe a PFS advantage of dacomitinib over afatinib in patients with or without brain metastases in subgroup analyses after PSM. We considered that this may be due to the small number of patients in the DC.

As pan-HER inhibitors, dacomitinib and afatinib share a spectrum of AEs, which mainly involve the skin and mucosa of the digestive tract [9,11,38]. In the ARCHER 1050 study, dacomitinib dose reductions were required in 66% of the patients because of intolerable AEs, and the most frequent AEs were diarrhea (86%), paronychia (61%), and rash (49%) [11]. We determined the dacomitinib dosage according to the patient’s comorbidities, weight, and physical status [16]. No patient required treatment cessation due to serious AEs, and only 15.9% (7/44) of the patients requested a dosage adjustment because of unbearable AEs. In this study, while the proportion of G1 AEs was significantly higher in the DC than in the AC (*p* = 0.006), the proportion of G3 diarrhea was significantly lower in the DC than in the AC (*p* = 0.036).

This study had several limitations. First, the AC and the DC used for internal exploration were small in scale relative to the ex-AC and came from only two hospitals within China, which may have led to selection bias. Second, the number of patients used to explore resistance mechanisms was too small, rendering the conclusions of the comparison between dacomitinib and afatinib underpowered. Considering the difficulty of conducting randomized, controlled clinical trials to study uncommon mutations, well-designed research is warranted to further clarify the optimal dosing of afatinib and dacomitinib, resistance mechanisms, and bridging issues of 2G TKIs with osimertinib, and we intend to collect more data to address these research questions in the future.

## 5. Conclusions

Dacomitinib demonstrated a more favorable efficacy than afatinib in terms of PFS and showed a manageable toxicity profile in patients with NSCLC carrying uncommon *EGFR* mutations. Differences in progression patterns and subsequent salvage therapy patterns were observed between the patients receiving dacomitinib and afatinib. The resistance mechanisms of 2G TKIs in this context remain to be explored.

## Figures and Tables

**Figure 1 cancers-14-05307-f001:**
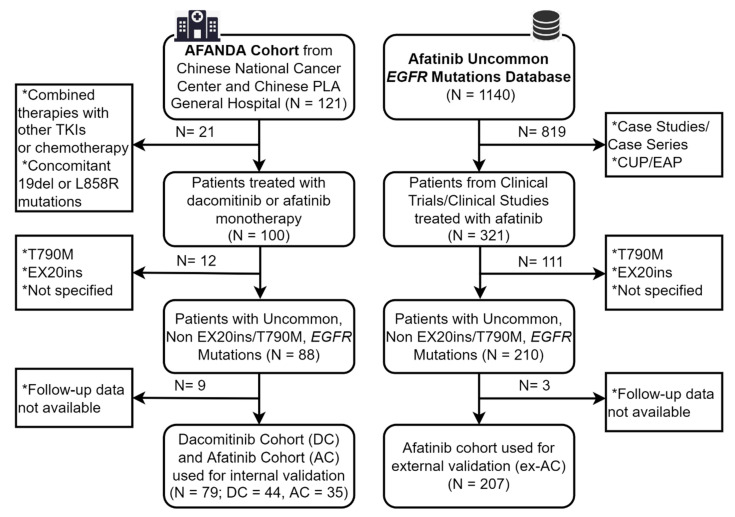
**Flow chart of patient selection.***EGFR*, epidermal growth factor receptor; TKI, tyrosine kinase inhibitor; EX20ins, exon 20 insertion mutations; DC, dacomitinib cohort; AC, afatinib cohort; ex-AC, external afatinib cohort; CUP/EAP, compassionate-use program/expanded-access program.

**Figure 2 cancers-14-05307-f002:**
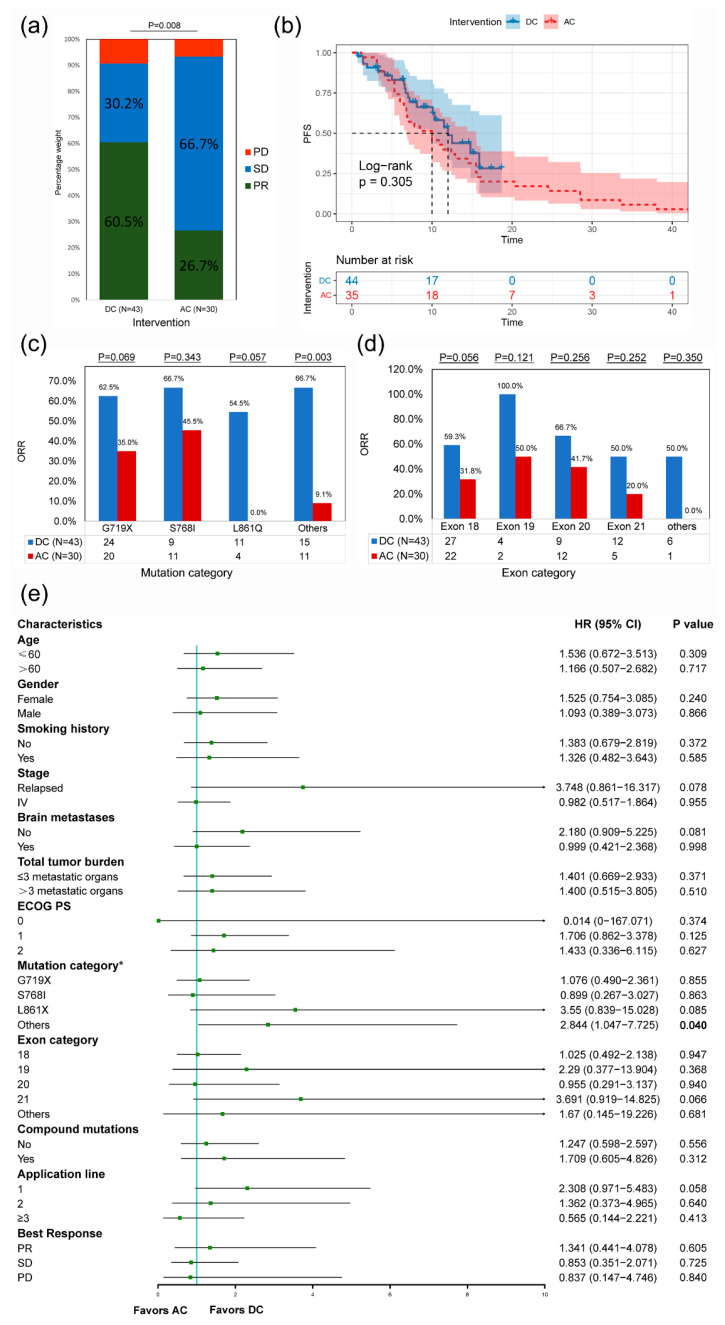
**Comprehensive comparison between the DC and the AC.** Stacked histogram of treatment responses (**a**), Kaplan-Meier curves (**b**), ORRs of different mutation categories (**c**), ORRs of different exon categories (**d**), and subgroup analysis of PFS (**e**) of the DC and the AC. ORR, objective response rate; PFS, progression-free survival; DC, dacomitinib cohort; AC, afatinib cohort; PR, partial response; SD, stable disease; PD, progressive disease; HR, hazard ratio; CI, confidence interval; ECOG PS, Eastern Cooperative Oncology Group performance status. * Uncommon mutation categories overlap with compound mutations, so each patient might belong to more than one group.

**Figure 3 cancers-14-05307-f003:**
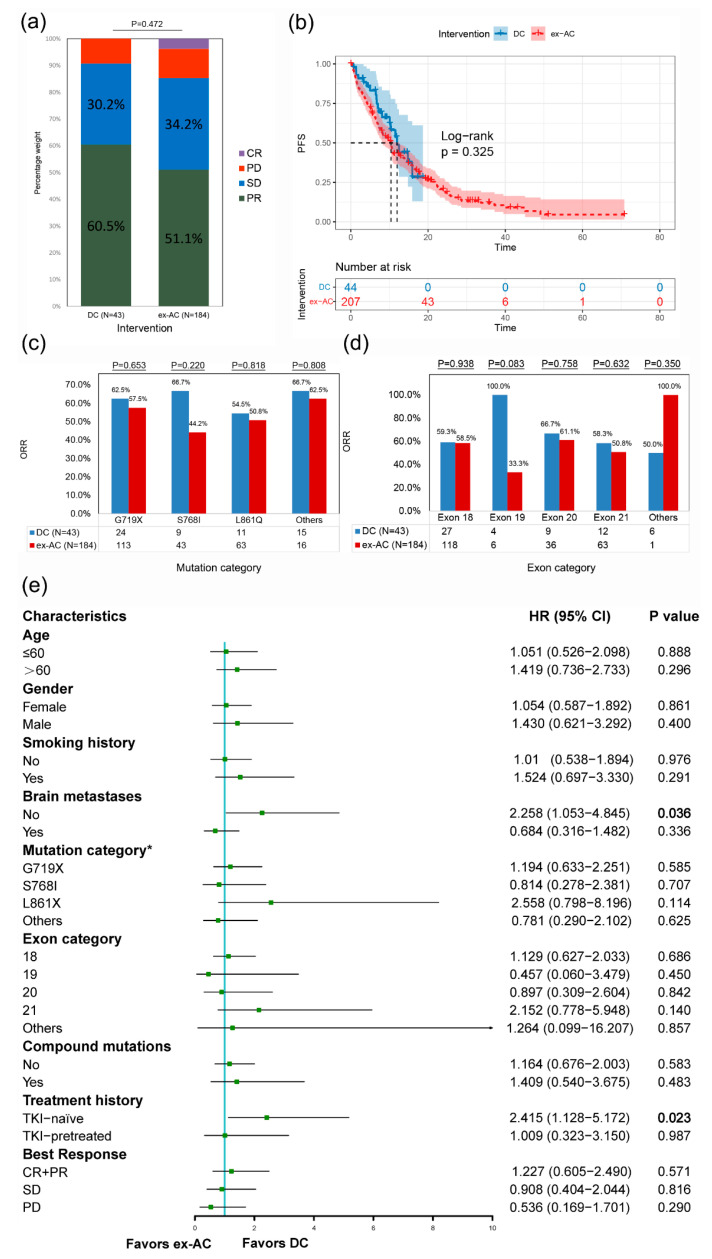
**Comprehensive comparison between the DC and the ex-AC before PSM.** Stacked histogram of treatment responses (**a**), Kaplan-Meier curves (**b**), ORRs of different mutation categories (**c**), ORRs of different exon categories (**d**), and subgroup analysis of PFS (**e**) of the DC and the ex-AC before PSM. ORR, objective response rate; PFS, progression-free survival; DC, dacomitinib cohort; AC, afatinib cohort; ex-AC, external afatinib cohort; CR, complete response; PR, partial response; SD, stable disease; PD, progressive disease; HR, hazard ratio; CI, confidence interval; TKI, tyrosine kinase inhibitor; PSM, propensity score matching. * Uncommon mutation categories overlap with compound mutations, so each patient might belong to more than one group.

**Figure 4 cancers-14-05307-f004:**
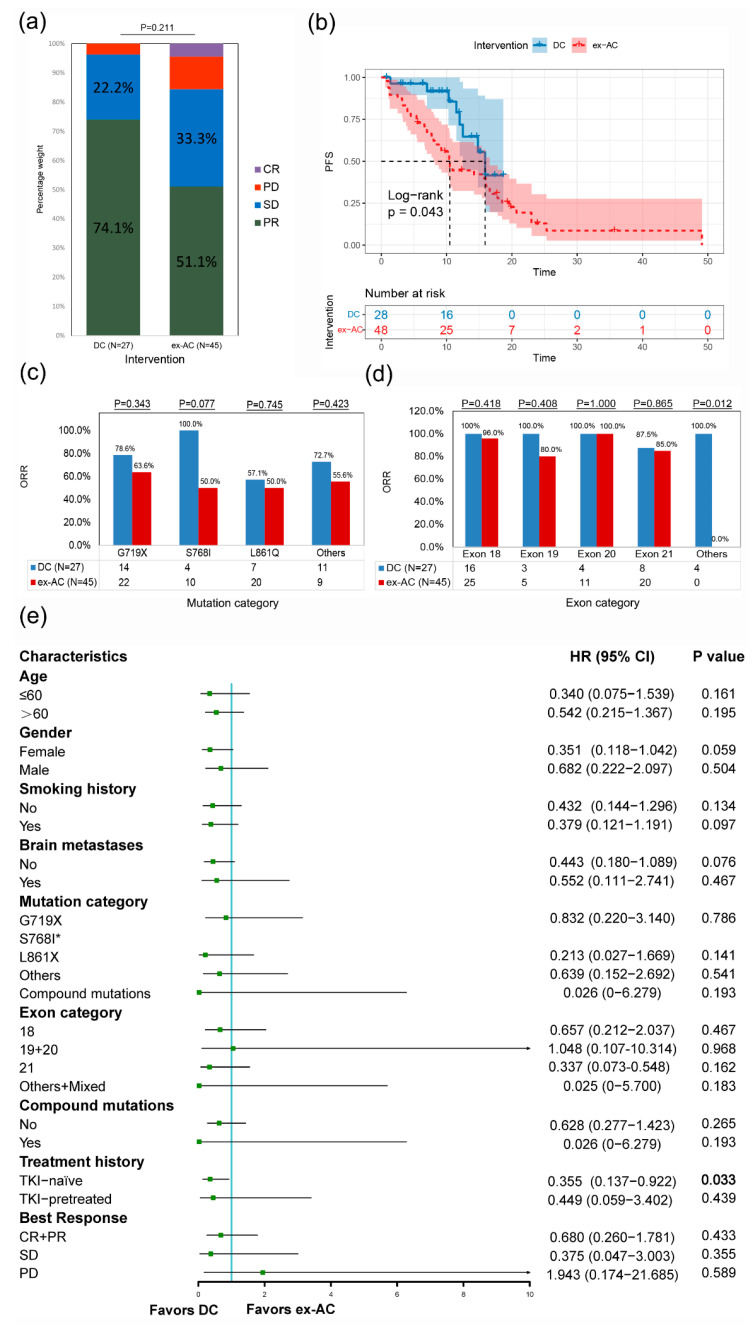
**Comprehensive comparison between the DC and the ex-AC after PSM.** Stacked histogram of treatment responses (**a**), Kaplan-Meier curves (**b**), ORRs of different mutation categories (**c**), ORRs of different exon categories (**d**), and subgroup analysis of PFS (**e**) of the DC and the ex-AC after PSM. ORR, objective response rate; PFS, progression-free survival; DC, dacomitinib cohort; AC, afatinib cohort; ex-AC, external afatinib cohort; CR, complete response; PR, partial response; SD, stable disease; PD, progressive disease; HR, hazard ratio; CI, confidence interval; TKI, tyrosine kinase inhibitor; PSM, propensity score matching. * Uncommon mutation categories overlap with compound mutations, so each patient might belong to more than one group. Since there were no cases with others in the ex-AC, the forest plot was not drawn.

**Figure 5 cancers-14-05307-f005:**
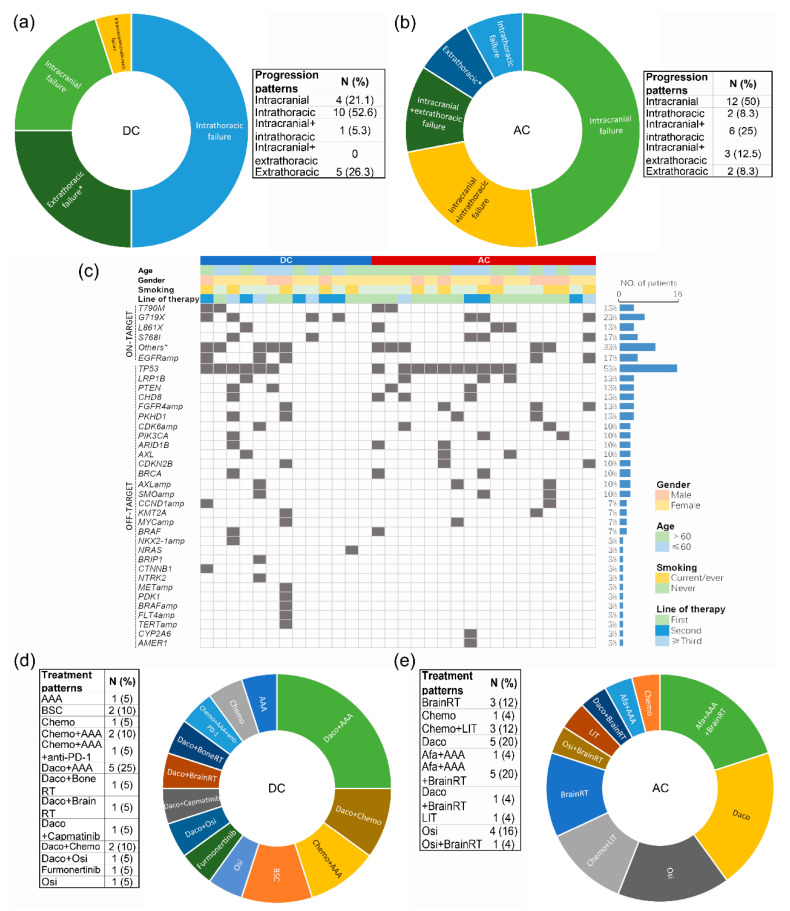
**Progression patterns (a,b), resistance mechanisms (c), and bridging therapies (d,e) in the DC and the AC.** AAA, anti-angiogenic agent; RT, radiotherapy; Chemo, chemotherapy; Daco, dacomitinib; Osi, osimertinib; LIT, local intervention therapy; PD-1, programmed cell death protein 1; amp, amplification; DC, dacomitinib cohort; AC, afatinib cohort. * Uncommon *EGFR* mutations other than major ones, T790M, and EX20ins. The non-*EGFR* mutations presented were all identified as resistance mechanisms rather than as present in the original tumor.

**Table 1 cancers-14-05307-t001:** Baseline characteristics of the DC and the AC (N = 79).

Characteristic	DC (N = 44)	AC (N = 35)	*p* Value
**Age**			0.859
≤60	21 (47.7)	16 (45.7)	
>60	23 (52.3)	19 (54.3)	
**Sex**			0.367
Female	33 (75.0)	23 (65.7)	
Male	11 (25.0)	12 (34.3)	
**Smoking history**			0.802
No	29 (65.9)	24 (68.6)	
Yes	15 (34.1)	11 (31.4)	
**Histology**			1.000
Adenocarcinoma	43 (97.7)	35 (100)	
Adenosquamous carcinoma	1 (2.3)	0	
**Stage**			0.286
Relapsed	12 (27.3)	6 (17.1)	
IV	32 (72.7)	29 (82.9)	
**Brain metastases**			0.125
No	24 (54.5)	25 (71.4)	
Yes	20 (45.5)	10 (28.6)	
**Total tumor burden**			0.758
<3 metastatic organs	34 (77.3)	26 (74.3)	
≥3 metastatic organs	10 (22.7)	9 (25.7)	
**ECOG PS**			0.683
0	11 (25.0)	6 (17.1)	
1	28 (63.6)	24 (68.6)	
2	5 (11.4)	5 (14.3)	
**Application line**	Median (range): 1 (1–6)	Median (range): 1 (1–5)	0.475
1	27 (61.4)	26 (74.3)	
2	9 (20.5)	5 (14.3)	
≥3	8 (18.2)	4 (11.4)	
**Mutation category ***			0.396
G719X	24 (54.5)	23 (65.7)	
G719X	9 (20.5)	10 (28.6)	
G719X + S768I	9 (20.5)	11 (31.4)	
G719X + L861X	2 (4.5)	0	
G719X + others	4 (9.1)	2 (5.7)	
S768I	9 (20.5)	12 (34.3)	
S768I + G719X	9 (20.5)	11 (31.4)	
S768I + others	0 (0)	1 (2.9)	
L861X	12 (27.3)	5 (14.3)	
L861X	9 (20.5)	3 (8.6)	
L861X + G719X	2 (4.5)	1 (2.9)	
L861X + others	1 (2.3)	1 (2.9)	
Others	15 (34.1)	11 (31.4)	
**Exon category ***			0.299
18	27 (61.4)	25 (71.4)	
19	4 (9.1)	3 (8.6)	
20	9 (20.5)	13 (37.1)	
21	13 (29.5)	6 (17.1)	
Others	6 (13.6)	1 (2.9)	
**Compound mutations**			0.274
No	28 (63.6)	18 (51.4)	
Yes	16 (36.4)	17 (48.6)	

Data are shown as n (%). ECOG PS, Eastern Cooperative Oncology Group performance status; DC, dacomitinib cohort; AC, afatinib cohort. * Uncommon mutation categories overlap with compound mutations, so each patient might belong to more than one group.

**Table 2 cancers-14-05307-t002:** Univariate and multivariate analyses of PFS in the pooled cohort of the DC and the AC (N = 79).

Characteristic	N	Univariate Analysis		Multivariate Analysis		
		Median (Months)	*p* Value	HR	95% CI	*p* Value
**Intervention**			0.305			**0.047**
Dacomitinib/Afatinib	44/35	12.0/10.0		1.909	1.009–3.610	
**Smoking history**			0.914			0.184
No/Yes	53/26	11.0/11.5		1.569	0.808–3.047	
**Brain metastases**			**0.004**			0.096
No/Yes	49/30	12.9/7.1		1.804	0.900–3.615	
**Total tumor burden**			**<0.001**			0.402
<3/≥3 metastatic organs	60/19	12.4/6.0		1.351	0.668–2.733	
**ECOG PS**			**<0.001**			**<0.001**
0	17	28.6		Reference	-	-
1	52	10.1		6.470	1.871–22.381	0.003
2	10	4.3		30.327	6.915–133.005	<0.001
**Application line**			**0.018**			**0.043**
1	53	12.9		Reference	-	-
2	14	8.1		2.16	0.985–4.737	0.055
≥3	12	6.5		2.55	1.073–6.059	0.034
**Mutation category ***			0.619			0.129
G719X	19	10.3		Reference	-	-
L861X	12	10.0		3.725	1.193–11.627	0.024
Others	16	12.4		1.958	0.756–5.071	0.166
Compound mutations	32	11.0		1.796	0.840–3.838	0.131

Data are shown as n (%). PFS, progression-free survival; ECOG PS, Eastern Cooperative Oncology Group performance status; DC, dacomitinib cohort; AC, afatinib cohort. Set variables before “/” as reference. * To meet the requirements of multivariate analysis, all patients were classified separately without repeated grouping. The S768I sub-category is not shown because there were no cases with single S768I in the combined cohort.

**Table 3 cancers-14-05307-t003:** Treatment-emergent AEs in the pooled cohort of the DC and the AC (N = 79).

AE	G1		*p* Value	G2		*p* Value	G3		*p* Value
	DC	AC		DC	AC		DC	AC	
Rash	22 (50.0)	15 (42.9)	**0.006**	16 (36.4)	3 (8.6)	0.142	2 (4.5)	0	**0.036**
Diarrhea	25 (56.8)	6 (17.1)		7 (15.9)	8 (22.9)		2 (4.5)	8 (22.9)	
Oral mucositis	16 (36.4)	12 (34.3)		9 (20.5)	8 (22.9)		2 (4.5)	0	
Paronychia	7 (15.9)	11 (31.4)		6 (13.6)	2 (5.7)		0	0	
Dry skin	12 (27.3)	1 (2.9)		5 (11.4)	1 (2.9)		1 (2.3)	0	
Others	4 (9.1)	0		1 (2.3)	0		0	0	

Data are presented as n (%). DC, dacomitinib cohort; AC, afatinib cohort; AE, adverse event. There were no grade 4–5 treatment-emergent AEs.

**Table 4 cancers-14-05307-t004:** Baseline characteristics of the DC and the ex-AC before and after PSM (1:2 matching).

Characteristic	Before PSM			After PSM ^#^		
	DC (N = 44)	ex-AC (N = 207)	*p* Value	DC (N = 28)	ex-AC (N = 48)	*p* Value
**Age**			0.173			0.839
<60	21 (47.7)	76 (36.7)		11 (39.3)	20 (41.7)	
≥60	23 (52.3)	131 (63.3)		17 (60.7)	28 (58.3)	
**Sex**			**0.005**			0.333
Female	33 (75.0)	107 (51.7)		20 (71.4)	29 (60.4)	
Male	11 (25.0)	100 (48.3)		8 (28.6)	19 (39.6)	
**Smoking history**			**<0.001**			0.150
No	29 (65.9)	74 (35.7)		18 (64.3)	27 (56.3)	
Yes	15 (34.1)	88 (42.5)		10 (35.7)	15 (31.3)	
Unknown	0	45 (21.7)		0	6 (12.5)	
**Brain metastases**			**<0.001**			0.951
No	24 (54.5)	186 (89.9)		22 (78.6)	38 (79.2)	
Yes	20 (45.5)	21 (10.1)		6 (21.4)	10 (20.8)	
**Treatment history**			**<0.001**			0.615
TKI-naïve	27 (61.4)	201 (97.1)		24 (85.7)	43 (89.6)	
TKI-pretreated	17 (38.6)	6 (2.9)		4 (14.3)	5 (10.4)	
**Mutation category ***			**0.001**			0.535
G719X	9 (20.5)	82 (39.6)		6 (21.4)	7 (14.6)	
L861X	9 (20.5)	57 (27.5)		6 (21.4)	13 (27.1)	
S768I	0	10 (4.8)		0	3 (6.3)	
Others	10 (22.7)	15 (7.2)		7 (25.0)	8 (16.7)	
Compound mutations	16 (36.4)	43 (20.8)		9 (32.1)	17 (35.4)	
**Exon category ***			**<0.001**			0.432
18	12 (27.3)	89 (43.0)		8 (28.6)	10 (20.8)	
19	4 (9.1)	4 (1.9)		3 (10.7)	4 (8.3)	
20	0	13 (6.3)		0	4 (8.3)	
21	11 (25)	60 (29.0)		8 (28.6)	13 (27.1)	
Others	2 (4.5)	0		1 (3.6)	0	
Mixed	15 (34.1)	41 (19.8)		8 (28.6)	17 (35.4)	
**Compound mutations**			**0.040**			0.772
No	28 (63.6)	162 (78.3)		19 (67.9)	31 (64.6)	
Yes	16 (36.4)	45 (21.7)		9 (32.1)	17 (35.4)	

Data are shown as n (%). DC, dacomitinib cohort; ex-AC, external afatinib cohort; TKI, tyrosine kinase inhibitor; PSM, propensity score matching. # Because some cases were not matched, the matching result was not an exact 1:2 match. * To meet the requirements of PSM analysis, all patients were classified separately without repeated grouping.

**Table 5 cancers-14-05307-t005:** Univariate and multivariate analyses of PFS in the pooled cohort of the DC and the ex-AC. (N = 251).

Characteristic	N	Univariate Analysis		Multivariate Analysis		
		Median (Months)	*p* Value	HR	95% CI	*p* Value
**Intervention**			0.325			**0.029**
Dacomitinib/Afatinib	44/207	12.0/10.4		1.953	1.071–3.563	
**Age**			0.569			0.553
<60/≥60	97/154	11.1/10.5		1.098	0.806–1.498	
**Sex**			**0.035**			0.056
Female/Male	140/111	12.0/8.2		1.397	0.991–1.97	
**Smoking history**			0.528			0.946
No	103	12.5		Reference	-	-
Yes	103	8.4		1.054	0.726–1.529	0.782
Unknown	45	10.5		1.063	0.691–1.635	0.781
**Brain metastases**			0.180			0.272
No/Yes	210/41	10.6/10.1		1.279	0.824–1.985	
**Treatment history**			**0.033**			**0.003**
TKI-naïve/pretreated	228/23	10.7/6.7		2.675	1.402–5.101	
**Mutation category ***			0.059			**0.040**
G719X	91	10.3		Reference	-	**-**
L861X	66	7.4		1.495	1.041–2.147	0.029
S768I	10	20.7		0.507	0.219–1.173	0.112
Others	25	10.7		1.320	0.755–2.308	0.329
Compound mutations	59	14.2		0.973	0.651–1.453	0.893

Data are shown as n (%). PFS, progression-free survival; DC, dacomitinib cohort; ex-AC, external afatinib cohort; TKI, tyrosine kinase inhibitor. Set variables before “/” as reference. * To meet the requirements of multivariate analysis, all patients were classified separately without repeated grouping.

## Data Availability

The dataset(s) supporting the conclusions of this article is(are) included within the article (and its additional file(s)).

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
