# Peer review of "Afatinib and Dacomitinib Efficacy, Safety, Progression Patterns, and Resistance Mechanisms in Patients with Non-Small Cell Lung Cancer Carrying Uncommon EGFR Mutations: A Comparative Cohort Study in China (AFANDA Study)"

_cancers, 2022, doi:10.3390/cancers14215307_

Round 1

Reviewer 1 Report

The authors are to be commended on undertaking a study in these rare EGFR mutation groups.  By the nature of the patient population and the treatments involved, prospective phase 3 studies remain impractical at the moment and hypothesis generating small, comparative, non-randomised ph2 trials are most likely to provide important clinical information, although the limitations of this approach are well understood and any conclusions have to be appropriately cautious, especially when making comparative claims

Methods: 
Need more information on the methodology and number of genes analysed (presume not whole genome?)
The last patient treated was April 30, 2022, and the study cut-off date was May 15, 2022 [Why such a short interval with no chance to analyse response or toxicity? Needs a comment in text]
Throughout manuscript: Consider reporting Male/Female as Sex not Gender [Gender could be reported as e.g. Man/Woman/Other - Minor comment but may be more important going forward]
Results:
Table 4 comes after Table 5 in text, which is confusing (I initially though it had been omitted)
Figure 5: Text in figure not clear - needs to enlarged/sharpened.
Need to clarify treatments in C and D e.g. AAA, antiangiogenesis therapy? Need to make it clear whether the non-EGFR gene mutations identified (e.g. TP53) could have been present in the original tumour rather than generated as a resistance mechanism.
Discussion: The authors state, '..we comprehensively compared 2G TKIs in patients with NSCLC carrying uncommon EGFR mutations with the aim to obtain valuable evidence for clinical decision-making...'  This statement is too strong based on the data presented. The authors themselves point out the small numbers in the DC group, so this can not be considered comprehensive or a definitive study for clinical decision making. 
General point:
The authors note that there may be racial confounding factors:  '...the population of internal exploration cohort s all Chinese, while that of external validation cohort is mainly non-Asian...'
In order to better 'validate' the comparison, the authors should have also compared the AC group with the ex-AC group and in particular with a PSM ex-AC group.  This would have tested/confirmed whether the response to AC was similar in both Chinese patients in the AFANDA study and in mixed/non-Asian patients extracted from the EGFR Mutations Database.  As it stands a major limitation of the current work is that the 'external' data only compares response to DC in Chinese pts with 'matched' non-Asian patients treated with AC.  It would be strong additional data to confirm that the response to AC was similar in both populations, and thereby give greater confidence in the conclusion increased efficacy of DC over AC.   

Author Response

Dear Reviewer 1:

Thank you for your comments concerning our manuscript entitled “Afatinib and dacomitinib efficacy, safety, progression patterns, and resistance mechanisms in patients with non-small cell lung cancer carrying uncommon EGFR mutations: a comparative cohort study in China (AFANDA study)” (ID: 1969058). Those comments are all valuable and very helpful for revising and improving our paper, as well as the important guiding significance to our researches. We have studied comments carefully and have made corrections which we hope meet with approval. Revised portions are marked in red in the paper using the “track changes” function. The main corrections in the paper and the response to your comments are as follows:

Question 1: Need more information on the methodology and number of genes analysed (presume not whole genome?)

Answer 1: Sorry for not providing enough information, we have added it in the appropriate part (page 3, section 2.4).

Question 2: The last patient treated was April 30, 2022, and the study cut-off date was May 15, 2022 [Why such a short interval with no chance to analyse response or toxicity? Needs a comment in text]

Answer 2: Thank you very much for your meticulous review of the manuscript. We apologize that we incorrectly wrote the inclusion deadline as 2022-4-30, when the last patient was actually included on 2022-4-15. This patient underwent a radiological review after 1 month of administration. We have corrected this in the original article.

Question 3: Throughout manuscript: Consider reporting Male/Female as Sex not Gender [Gender could be reported as e.g. Man/Woman/Other - Minor comment but may be more important going forward]

Answer 3: Thank you for your careful suggestions, and we have revised this mistake as suggested.

Question 4: Table 4 comes after Table 5 in text, which is confusing (I initially though it had been omitted)

Answer 4: We are very sorry that the order of the tables was wrong due to a typographical problem, and we have corrected it.

Question 5: Figure 5: Text in figure not clear - needs to enlarged/sharpened.

Answer 5: Thank you for your suggestion. We have enlarged this figure as indicated.

Question 6: Need to clarify treatments in C and D e.g. AAA, antiangiogenesis therapy?

Answer 6: Sorry for this miss. We have clarified these abbreviations in the main text and figure legends.

Question 7: Need to make it clear whether the non-EGFR gene mutations identified (e.g. TP53) could have been present in the original tumour rather than generated as a resistance mechanism.

Answer 7: Sorry for not explaining this issue in the manuscript. The genes listed in Figure 5 are all secondary genetic alterations that occur after drug resistance. We have added the relevant description in the figure legend of Figure 5 as “the non-EGFR mutations presented were all identified as resistance mechanisms rather than presented in the original tumor.".

Question 8: The authors state, '..we comprehensively compared 2G TKIs in patients with NSCLC carrying uncommon EGFR mutations with the aim to obtain valuable evidence for clinical decision-making...'  This statement is too strong based on the data presented. The authors themselves point out the small numbers in the DC group, so this can not be considered comprehensive or a definitive study for clinical decision making. 

Answer 8: Thank you for your suggestion! We have revised the above statement as you suggested as "In the current study, we tentatively compared 2G TKIs in patients with NSCLC carrying uncommon EGFR mutations with the aim to obtain valuable evidence for clinical reference.".

Question 9: In order to better 'validate' the comparison, the authors should have also compared the AC group with the ex-AC group and in particular with a PSM ex-AC group.  This would have tested/confirmed whether the response to AC was similar in both Chinese patients in the AFANDA study and in mixed/non-Asian patients extracted from the EGFR Mutations Database. 

Answer 9: Thank you very much for your insightful and constructive advice! In fact, as we mentioned in the manuscript, there is a heterogeneity between the AC and ex-AC. AC patients are mainly derived from real-world settings and are all Chinese. The ex-ac cohort, on the other hand, is composed of patients from clinical trials screened in the database, and is predominantly of non-Asian origin. Because of this, in order to avoid bias, we did not combine AC and ex-AC in the design of the study for analysis, but instead used the two cohorts as an internal exploration cohort and an external validation cohort, separately. In other words, if the same conclusion was reached in both cohorts containing different ethnic groups and different treatment settings, this indicates that our results are more robust and convincing. And indeed, our study yielded consistent results in the multivariate analyses for all three comparisons (DC vs AC, DC vs ex-AC before PSM, and DC vs ex-AC after PSM). Comparing AC and ex-AC, however, may be out of line with the trial design and, because of differences in ethnicity and treatment setting, would lead to less convincing conclusions that would be confusing and difficult for readers to understand. Besides, we are also sorry that the description in the limitation section was somewhat misleading. We have changed the original description in the limitation part as "First, the AC and DC used for internal exploration were small-scaled relative to the ex-AC and came from only two hospitals within China, which may have been subject to selection bias".

We tried our best to improve the manuscript and made some changes  to the manuscript. These changes will not influence the content and framework of the paper. And here we did not list the changes but marked in red or blue in the revised paper (including tables and pictures).

We appreciate for your warm work earnestly and hope that the correction will meet with approval.

Once again, thank you very much for your comments and suggestions!

Reviewer 2 Report

This manuscript compared the efficacy, safety, progression patterns, and resistance mechanisms dacomitinib and afatinib in patients with NSCLC carrying uncommon EGFR mutations. This study is well designed and organized. It reported important data about the treatment of NSCLC and provided evidence for clinical decision-making. Overall, this is a high quality study and is interesting to readers. My only question to this study is that why the data from AFANDA cohort and ex-AC cohort was analyzed separately. Can they be combined in analysis?

Author Response

Dear Reviewer 1:

Thank you for your comments concerning our manuscript entitled “Afatinib and dacomitinib efficacy, safety, progression patterns, and resistance mechanisms in patients with non-small cell lung cancer carrying uncommon EGFR mutations: a comparative cohort study in China (AFANDA study)” (ID: 1969058). Those comments are all valuable and very helpful for revising and improving our paper, as well as the important guiding significance to our researches. We have studied comments carefully and have made corrections which we hope meet with approval. Revised portions are marked in red in the paper using the “track changes” function. The main corrections in the paper and the response to your comments are as follows:

Question 1: My only question to this study is that why the data from AFANDA cohort and ex-AC cohort was analyzed separately. Can they be combined in analysis?

Answer 1: First of all, thank you very much for your recognition of our work! In fact, we did not combine AC and ex-AC because of the heterogeneity of the two cohorts. patients in AC were mainly from real-world settings and were all Chinese. In contrast, the ex-AC patients were all derived from clinical trial settings screened in the database and were mainly of non-Asian origin. Considering the potential significant bias after combining, we discarded the combined analysis. To make full use of the data, we used the two cohorts as an internal exploration cohort and an external validation cohort respectively, which avoided the potential bias from the combined analysis and also achieved the effect of mutual validation. In other words, if we obtained the same conclusion in both cohorts containing different ethnic groups and different treatment settings, this indicates that our results are more robust and convincing. 

We tried our best to improve the manuscript and made some changes  to the manuscript. These changes will not influence the content and framework of the paper. And here we did not list the changes but marked in red or blue in the revised paper (including tables and pictures).

We appreciate for your warm work earnestly and hope that the correction will meet with approval.

Once again, thank you very much for your comments and suggestions!